# Assessing the Implementation of Pharmacogenomic Panel-Testing in Primary Care in the Netherlands Utilizing a Theoretical Framework

**DOI:** 10.3390/jcm9030814

**Published:** 2020-03-17

**Authors:** Cathelijne H. van der Wouden, Ellen Paasman, Martina Teichert, Matty R. Crone, Henk-Jan Guchelaar, Jesse J. Swen

**Affiliations:** 1Department of Clinical Pharmacy & Toxicology, Leiden University Medical Center, 2333 ZA Leiden, The Netherlands; 2Leiden Network for Personalised Therapeutics, Leiden University Medical Center, 2333 ZA Leiden, The Netherlands; 3Community Pharmacy De Klipper, 2692 AH ‘s Gravenzande, The Netherlands; 4Department of Public Health and Primary Care, Leiden University Medical Center, 2333 ZA Leiden, The Netherlands

**Keywords:** pharmacogenomics, implementation, qualitative, framework, pharmacist, panel-testing

## Abstract

Despite overcoming many implementation barriers, pharmacogenomic (PGx) panel-testing is not routine practice in the Netherlands. Therefore, we aim to study pharmacists’ perceived enablers and barriers for PGx panel-testing among pharmacists participating in a PGx implementation study. Here, pharmacists identify primary care patients, initiating one of 39 drugs with a Dutch Pharmacogenetic Working Group (DPWG) recommendation and subsequently utilizing the results of a 12 gene PGx panel test to guide dose and drug selection. Pharmacists were invited for a general survey and a semi-structured interview based on the Tailored Implementation for Chronic Diseases (TICD) framework, aiming to identify implementation enablers and barriers, if they had managed at least two patients with actionable PGx results. In total, 15 semi-structured interviews were performed before saturation point was reached. Of these, five barrier themes emerged: (1) unclear procedures, (2) undetermined reimbursement for PGx test and consult, (3) insufficient evidence of clinical utility for PGx panel-testing, (4) infrastructure inefficiencies, and (5) HCP PGx knowledge and awareness; and two enabler themes: (1) pharmacist perceived role in delivering PGx, and (2) believed clinical utility of PGx. Despite a strong belief in the beneficial effects of PGx, pharmacists’ barriers remain, an these hinder implementation in primary care.

## 1. Introduction

Pharmacogenomics (PGx) informed prescribing is one of the first applications of genomics in medicine [1,2]. It promises to personalize pharmacotherapy by using an individual’s germline genetic profile to guide optimal drug and dose selection [3,4]. This removes the traditional ‘trial and error’ approach of drug prescribing, thereby promising safer and more (cost-) effective drug treatment [5,6]. Over a decade ago, we analysed the implementation barriers preventing the wide-spread adoption of PGx testing to guide dose and drug selection [7]. Since then, many barriers have been overcome, including the generation of evidence supporting the clinical utility of single gene-drug interactions [8,9,10,11], the development of clinical guidelines based on systematic review of literature [12,13,14], the selection of clinically relevant PGx tests [15] and the stimulation of healthcare professional’s (HCP) belief in PGx guided prescribing [16,17].

Despite overcoming these barriers and the initiation of numerous implementation programs [18,19], PGx is still not widely implemented in primary care. Remaining barriers outlined in the literature include a lack of evidence supporting clinical utility, undetermined timing and methodology of testing, a lack of HCPs’ confidence to apply it in practice, patients’ concern for privacy issues and a lack of reimbursement [20,21,22,23]. Furthermore, remaining barriers may differ when delivered in a single-gene reactive approach or a pre-emptive panel approach. Currently identified barriers have been reported in studies investigating HCPs with little to no experience with PGx testing. However, studies among HCPs who applied PGx may provide a more accurate view of implementation barriers. Lemke et al. and Unertl et al. investigated barriers experienced by physicians who had applied pre-emptive PGx-panel-testing in practice within study settings [24,25]. These studies reported several challenges in implementation, including difficulties in interpreting the PGx test results despite the availability of guidelines, delays in receiving the results, lack of time to adequately inform their patients, and concerns about responsibility for the PGx results [24,25].

As well recognized drug experts and designated HCPs to handle medication surveillance, pharmacists may be better-suited candidates for applying PGx testing than physicians [26,27,28]. A pharmacist-initiated approach to PGx has been proven feasible and is considered promising for the implementation of PGx testing in primary care [28]. In such an approach, pharmacists are responsible for selecting patients eligible for PGx testing and in utilizing their results to guide dose and drug selection at dispensing. Although some reported implementation barriers and enablers may be comparable among pharmacists and physicians, some may be specific to pharmacists, due to different settings, roles, and knowledge. However, implementation barriers and enablers encountered by community pharmacists who have hands-on experience with PGx testing are yet undetermined. Identification of these barriers may further help selecting and tailoring implementation strategies to stimulate widespread adoption. Over the years, the implementation science field has called for the more explicit use of theoretical frameworks to investigate implementation barriers and facilitators [29,30]. However, within the field of PGx, a formal qualitative study using such frameworks has not yet been performed. Therefore, we studied pharmacists’ perceived remaining barriers, preventing and enablers facilitating implementation of pharmacist-initiated PGx panel-testing in primary care utilizing mixed-methods, including qualitative investigation using theoretical frameworks.

## 2. Materials and Methods

Pharmacist perceived implementation enablers and barriers of PGx testing were studied among community pharmacists in the Netherlands who have applied PGx testing within the PREPARE (PREemptive Pharmacogenomic Testing for Preventing Adverse Drug Reactions) study of the Ubiquitous Pharmacogenomics Consortium (U-PGx). U-PGx utilizes a multi-component approach to implement PGx which has previously been published elsewhere [19].

### 2.1. Study Setting

In this study, pharmacists involved with enrolment for the PREPARE study are investigated. Enrolment for the PGx intervention arm was initiated in October 2018. In brief, pharmacists select and enrol patients under their care, who plan to initiate one of 39 drugs with a Dutch Pharmacogenetics Working Group (DPWG) recommendation (see Appendix A). After informed consent, a DNA saliva sample is collected in the pharmacy and sent to Leiden University Medical Center Pharmacogenetics Lab for a PGx panel test, encompassing 12 pharmacogenes (see Appendix A). This panel differs from a previously published version of the panel in the PREPARE study [15], since it excludes variants in *HLA-A* and *NUDT15*. Actionable test results for the drug of enrolment are directly communicated to the pharmacist by phone and email. It is left to the discretion of the pharmacist if and how to use the DPWG recommendation, whether to discuss results with the treating physician and how to report the results to patients. Both pharmacists and physicians are free to choose whether or not to adhere to the DPWG recommendation. Other PGx test results are communicated by email. The pharmacist is responsible for recording the PGx panel results in the electronic medical record (EMR), to enable medication surveillance for future prescriptions through automated pop-ups of the relevant DPWG recommendation. As part of the U-PGx implementation strategy, enrolling pharmacists were provided with a flexible e-learning program to educate them on using PGx before enrolling patients [31]. Many participating pharmacists previously gained experience with PGx by participating in a PGx pilot study, preceding the current study [28,32].

In the Dutch healthcare system, patients are typically listed with one general practitioner (GP) and one community pharmacy. More specialized care is provided in outpatient or hospital settings by medical specialists. Pharmacists’ level of expertise generally increases from community pharmacist specialist in training to managing pharmacist. Pharmacies may be independent, part of a franchise or part of a pharmacy group. Additionally, a pharmacy may be located in a dedicated health center, often sharing a building with multiple healthcare providers, such as GPs and physiotherapists. Pharmacists maintain an EMR containing dispensing history, relevant contraindications, and drug allergies. EMRs are used by all pharmacists for medication surveillance at the time of drug dispensing. Pharmacists generally delegate specific medication surveillance tasks to the pharmacy technicians. Within the U-PGx project, pharmacists receive a report holding PGx panel results for 12 pharmacogenes for enrolled patients by secured e-mail. It is the pharmacist’s responsibility to record these 12 phenotypes corresponding to 12 genes as contra-indications in the EMR, to enable medication surveillance. Although the term contra-indication generally indicates a particular drug should not be used, this is not implied in the PGx scenario. The term contra-indication only relates to the format in which the genotype predicted phenotypes should be recorded in the EMR, to trigger pop-up messages with PGx information during drug prescribing and dispensing. Pharmacists are encouraged to record all results, regardless of the phenotype being aberrant or normal, to document that testing for that particular gene has been performed. This has to be done manually, per individual phenotype. The DPWG recommendations are incorporated in the nation-wide clinical decision support system (CDSS), through the G-standard, and therefore enable medication surveillance using these results in future prescriptions [33]. This enables a pop-up of the relevant DPWG recommendation when an actionable drug-gene interaction is encountered. As a result of an actionable gene-drug interaction, pharmacists can advise dose or therapy changes to the treating physicians. Physicians must formally approve all pharmacotherapy changes. However, to facilitate timely drug dispensing, pharmacists and GPs commonly make preemptive agreements, allowing pharmacists to make changes for the most prevalent situations.

By law, all citizens of the Netherlands must have basic healthcare insurance. PGx testing costs to investigate the cause of an ADR are reimbursed. Additionally, several healthcare insurers offer PGx screening as part of an optional reimbursement package.

### 2.2. Study Design

The primary aim of our study was to identify pharmacists’ perceived remaining barriers preventing and enablers facilitating implementation of pharmacist-initiated PGx in primary care. Combined mixed-methods using both surveys and semi-structured interviews, based on theoretical frameworks, were used to assess the primary study aim. Firstly, to investigate shared decision making, report of results to patients, and time allocation, PGx recommendation specific surveys were collected following report of every individual actionable PGx recommendation to pharmacists. Secondly, to identify remaining implementation barriers and enablers, both a semi-structured interview and pharmacist specific survey, regarding demographics and attitudes towards PGx, were performed. Pharmacists were invited for a semi-structured interview and general survey if they had managed at least two patients with actionable PGx results. A detailed pharmacist journey, per enrolled patient, and data collection logistics are outlined in Figure 1.

### 2.3. Ethical Approval

All pharmacists participating in the semi-structured interviews and pharmacist specific survey provided written informed consent before participation. This study was conducted in accordance with the Declaration of Helsinki, and the protocol of this was approved by the Ethics Committee of Leiden University Medical Center (LUMC) as an addendum to the PREPARE Study (NL60069.058.16).

### 2.4. Data Collection: PGx Recommendation Specific Surveys

A PGx recommendation specific survey was conducted following the report of every individual actionable PGx recommendation to pharmacists (see Appendix A). These surveys assess whether the DPWG recommendation was discussed with the treating physician, whether the pharmacist supports the agreed upon pharmacotherapy adjustment, and asks who reported the results to the patient, and how much time was spent handling the recommendation.

### 2.5. Data collection: Pharmacist Specific Survey and Semi-Structured Interviews among Participating Pharmacists

The pharmacist specific survey consisted of pharmacist and pharmacy demographics, and pharmacist self-reported knowledge and perceptions of PGx (see Appendix A). For the interviews, an initial interview template was constructed using components from the Tailored Implementation for Chronic Diseases (TICD) and two other frameworks [34,35,36,37]. The TICD includes 57 individual determinants grouped in 7 domains (guideline factors, individual health professional factors, patient factors, professional interactions, incentives and resources, capacity for organizational change; social, political, and legal factors), and links individual determinants with one or more of 116 behavioral interventions [34]. The initial template was further adapted using literature on PGx implementation and current knowledge about the pharmacists’ role in PGx implementation. The initial template used to conduct the first interview is displayed in Appendix A. Interviews regarded pharmacists’ experience with PGx, both in and outside the context of this study. Interviews were held in the pharmacy of the interviewed pharmacist and performed in Dutch. The first three interviews were conducted by two researchers (CW and EP) together, to ensure similar approaches for subsequent interviews performed by a single interviewer. The audio of the interviews was recorded and transcribed with Microsoft Word. Participant identifying information was removed. The transcript was initially coded in Microsoft Word by two researchers independently (CW and EP) and subsequently discussed in order to identify novel domains. The initial interview template was edited after subsequent interviews to include novel agreed-upon domains. New interviews were conducted, until no novel domains emerged and theoretical saturation was reached.

### 2.6. Data Analysis

For the survey responses, frequency statistics were calculated using IBM SPSS Statistics version 25 (IBM, New York, NY, USA). These were used to contextualize the qualitative results. For the interviews, ATLAS.ti version 7.5.18 (ATLAS.ti GmbH, Berlin, Germany) was used to electronically code and manage qualitative data, and to generate reports of coded text for analysis. Researchers inductively agreed that the TICD framework best fit the determinants of implementation to guide qualitative data analysis. Therefore, the finalized coding scheme was mapped and structured according to the TICD framework [34,35]. Salient themes were identified from this framework for final data analysis. Illustrative quotes were derived from the semi-structured interviews and were translated into English.

## 3. Results

### 3.1. Participant Demographics

Overall, 19 pharmacists managed at least 2 actionable PGx results and were approached for a semi-structured interview and pharmacist specific interview. Of these, 15 pharmacists were interviewed before saturation was reached. See Table 1 for the demographics of interviewed pharmacists, their pharmacies and drugs on which patients with actionable drug-gene interactions were enrolled. Interviewed pharmacists were on average 38.5 years old (range: 25–59) and had 12.9 years of work experience (range: 0.5–30). The mean self-reported PGx knowledge was 3.3 out of 5 (range: 2–4) and mean reported belief in the effect of pre-emptive PGx testing was 4.1 (range: 3–5). Furthermore, 33.3% of pharmacists reported to have either partially or fully completed the offered the U-PGx e-learning on PGx. Over half of the actionable gene–drug pairs for which pharmacists received recommendations concerned *SLCO1B1*-simvastatin and *CYP2D6*-metoprolol. The mean duration of the interviews was 34 min (range: 19–50 min).

### 3.2. PGx Recommendation Specific Surveys

PGx recommendation specific surveys were collected, concerning 92 actionable PGx reports. See Table 2 for an overview of shared decision making, report of results to patients, and time allocation. In 77.2% of cases, pharmacists discussed the PGx results with the treating physicians. The majority (56.5%) of PGx results were reported to patients by the pharmacists themselves. On average, pharmacists reported spending a total of 18 min in handling the recommendation.

### 3.3. Interview Findings

To analyze qualitative data, the final codes were mapped to 40 determinants of the TICD framework, within all 7 domains (guideline factors, individual health professional factors, patient factors, professional interactions, incentives and resources, social, political and legal factors and capacity for organizational change). Regarding reported barriers, five salient themes emerged from the analysis: (1) unclear procedures outside the study, (2) undetermined reimbursement for testing and consulting, (3) insufficient evidence of clinical utility for PGx panel-testing, (4) infrastructure inefficiencies and (5) HCP PGx knowledge and awareness. Regarding implementation enablers, two salient themes emerged from the analysis: (1) pharmacist perceived role in delivering PGx, and (2) believed effects of PGx. These are further presented per congruent TICD domain in the following sections. Illustrative quotes for barriers are presented in Table 3 and for enablers in Table 4.

#### 3.3.1. Interview Findings: Pharmacist Perceived Remaining Barriers

##### Unclear Procedures Outside Study Setting

The following barriers correspond to the TICD Individual health professional factors domain. The majority of pharmacists reported that they were not sure on how to request PGx testing outside the research-setting. More specifically, they were not sure if test costs would be reimbursed by the insurance companies if requested by a pharmacist, when to request testing (diagnostically or pre-emptively), how to request testing, at which laboratory to request testing, and unsure about which genes to test or whether to test the entire PGx panel. One pharmacist reported being aware of how to request a test outside the study setting, because of being involved in a clinical project. Several pharmacists reported that for application outside the study setting a set of practical guidelines specifying which patients and genes to test, which is the responsible HCP for requesting a test and reimbursement procedures, would be very helpful.

The following barrier corresponds to the TICD Capacity for organizational change domain. A number of pharmacists expressed that they felt the assistance of pharmacist professional organizations would be very helpful in creating both practical implementation guidelines, to clarify PGx testing logistics, and information brochures to inform patients regarding PGx. Some pharmacists felt strongly that the pharmacist professional organization should publish a PGx policy statement and coordinate implementation centrally. One pharmacist suggested that this policy statement could help to advocate for the reimbursement of PGx testing.

##### Undetermined Reimbursement for Testing and Consulting

The following barrier corresponds to the TICD Incentives and resources domain. All pharmacists reported the lack of reimbursement of the actual PGx test and lack of reimbursement of the time spent by pharmacists to record and act upon the PGx recommendation to be major implementation barriers.

Regarding the costs of the actual PGx test, many pharmacists felt strongly that the test costs were currently too high. The reimbursement status of PGx testing was unclear for the majority of pharmacists. Some believed it was covered by deductible expenses or that reimbursement was included in healthcare insurer’s optional insurance packages, while others believed that the patient had to pay out of pocket for testing. For some pharmacists, this uncertainty was a reason not to request testing outside the study setting, as they did not want to risk unplanned costs for the patient.

Pharmacists also felt strongly that the time they spent on recording and interpreting PGx test results should be reimbursed. One pharmacist noted that they did not mind the lack of reimbursement in a study-setting. Yet, reimbursement for pharmacist time would be imperative for routine implementation. Another pharmacist suggested a possible reimbursement route to be one comparable to the available reimbursement of medication reviews.

The following barrier corresponds to the TICD Social, political and legal factors domain. Many pharmacists stated a number of approaches in acquiring reimbursement for PGx testing. These approaches included advocacy from professional organizations, generation of convincing evidence for patient benefit and cost-effectiveness of a PGx panel test, a decrease in PGx testing costs, and increased PGx testing in routine care. One pharmacist compared reimbursement of PGx testing to blood group typing, suggesting that as PGx testing becomes more routine and common, insurers will reimburse it over time.

##### Insufficient Evidence of Clinical Utility for PGx Panel-Testing

The following barrier corresponds to the TICD Guideline factors domain. Pharmacists reported insufficient evidence for the clinical utility of a panel-approach outside of the study setting. Similar to the requirement for novel drugs to demonstrate a favorable benefit/risk ratio and cost-effectiveness, they also felt this is a strong requirement for the implementation of PGx panel-testing. In contrast, specific drug-gene interactions were deemed to be supported by convincing evidence by some pharmacists. For example, multiple pharmacists reported that they would support the pre-emptive testing of CYP2C19 in all patients initiating clopidogrel in their practice. Reasons stated for prioritizing this particular gene–drug pair were the relatively high prevalence of patients with aberrant genotypes and the strong evidence for patient benefit. However, practical constraints were preventing them from implementation. Additional drugs, for which PGx interactions were deemed important for pharmacists to implement, were statins and antidepressants, the reasons for this being that there was sufficient evidence and perceived patient benefit.

##### Infrastructural Inefficiencies

The following barrier corresponds to the TICD Incentives and resources domain. Pharmacists considered the management of gene-drug interactions as a routine part of medication surveillance. They felt responsible for the recording of PGx results and the integration within their clinical workflow, to the same standards as that of acting upon other drug-drug interactions. Overall, pharmacists were not satisfied with the performance of the CDSS. Primarily, they reported that recording the test results for the 12 pharmacogenes in the EMR was time-consuming and error-prone. Moreover, a widely used computerized pharmacy system only supported the recording of 10 contra-indications per patient. As a result, pharmacists were unable to record the 12 reported phenotypes as contra-indications and therefore information was lost. All pharmacists considered it their responsibility to record the results, as opposed to delegating it to pharmacy technicians. Additionally, the majority of pharmacists incorporated a quality check by a second pharmacist to avoid recording erroneous results. This was especially important to them because they noted that genetic test results persist throughout life. One pharmacist suggested a perfect IT system would automatically import the results from the performing laboratory, for example utilizing a nation-wide EMR sharing infrastructure.

The following barrier corresponds to the TICD Guideline factors domain. The DPWG recommendations were considered clear and easily interpretable overall. However, a number of reported barriers were associated with clarity and interpretability of the SLCO1B1-statin and CYP2D6-metoprolol recommendations. A number of pharmacists reported DPWG recommendations, which they found unconcise and unclear. For example, the DPWG recommendation for the CYP2D6 poor metabolizer-metoprolol interaction has two potential actions which also depend on the indication of metoprolol use and the symptoms the patient may have experienced. As a result of this perceived unclarity, pharmacists reported to be less confident in discussing the results with the treating physician and less likely to adhere to the recommendation.

##### HCP PGx Knowledge and Awareness

The following barrier corresponds to the TICD Professional interactions domain. Pharmacists regarded themselves at a reasonable level of knowledge about PGx. They reported their PGx knowledge to obtained through both personal interest and participation in this implementation study. However, they noted that colleague pharmacists, GPs and medical specialists, who were not involved in a PGx study, had very little awareness and knowledge of PGx. Lack of knowledge of colleagues involved in the healthcare chain was often reported as a prominent implementation barrier. In particular cases, the inequality in PGx knowledge between the enrolling pharmacists and the treating physician hampered shared decision making and adherence to the DPWG recommendation. Pharmacists, however, did note a diversity in knowledge across medical specialists. Overall, pharmacists perceived the PGx knowledge of GPs to be lower than their own. A minority of GPs was reported to be knowledgeable and was able to request PGx tests.

The following barrier corresponds to the TICD Individual health professional factors domain. In addition, pharmacists reported the lack of awareness of the possibility of PGx testing among the general population of pharmacists, physicians, and patients as being an important barrier. To stimulate awareness among pharmacists, many suggested more publications on PGx in the professional journal of Dutch pharmacists. To stimulate awareness among patients, some pharmacists proposed generating more media attention for PGx testing. Other pharmacists stimulated the initiation of pharmacotherapy audit meetings with their colleague GPs, to educate them on PGx and create awareness within the GP community. One pharmacist underlined the importance of sharing PGx success stories, for example of patients for whom PGx testing contributed to improved outcomes.

#### 3.3.2. Interview Findings: Pharmacist Perceived Enablers

##### Perceived Role in Delivering PGx

The following enabler corresponds to the TICD Professional interactions domain. Pharmacists generally agreed that they were primary and leading candidates for implementing PGx in routine care; feeling responsible for patient selection, requesting PGx tests, recording the PGx results in the EMR, acting upon gene–drug interactions by discussing with the treating physician, and finally to report and explain the agreed-upon pharmacotherapy adjustment to the patient. Particularly, acting upon PGx testing results was a task that pharmacists felt very capable of doing, as they reported being experts in resolving drug interactions in medication surveillance. They additionally noted that they felt GPs did not have time for this additional task and that GPs expect this to be the pharmacist’s expertise. Most pharmacists, however, felt that following up on patient symptoms is a shared responsibility with the GP.

The following enabler corresponds to the TICD Incentives and resources domain. Generally, pharmacists found handling PGx results enjoyable and felt appreciated for their work, both by GPs and by patients. A few pharmacists were extremely positive and noted that their added value by successful and beneficial reporting of PGx results to patients was the reason why they were in their profession. Additionally, the majority of pharmacists agreed that being the expert in PGx was of strategic value for the pharmacist profession, since it is a clear and concise example of how pharmacists contribute to healthcare through medication surveillance.

The following enabler corresponds to the TICD Capacity for organizational change and Social, political and legal factors domains. Pharmacists also felt that they had the capable leadership skills required for implementing PGx; being confident in their knowledge and ability to perform all tasks in the implementation chain. Interestingly, one pharmacist reported that they had been influenced by another pharmacist, who had taken the initiative to test all patients for CYP2C19 initiating clopidogrel in his practice. This influenced this particular pharmacist to initiate a similar initiative, not only in his practice but within all pharmacies in a formal regional collaboration.

##### Believed Effects of PGx

The following enabler corresponds to the TICD Individual health professional factors domain. Even in the absence of high-grade evidence for pre-emptive PGx panel-testing, all pharmacists reported to strongly believe in the beneficial effects of PGx guided pharmacotherapy on a number of domains: pharmacotherapy, pharmacist added value and knowledge, and professional interactions.

When delivered in a pre-emptive setting, pharmacists reported that they believed that PGx guided pharmacotherapy would be particularly beneficial for identifying patients who are at higher risk for adverse drug reactions. A number of beneficial downstream effects were reported by pharmacists: improvement of drug adherence, prevention of hospital admissions, reduction of trial and error in finding the correct dose, and time-saving. When delivered in a diagnostic setting, pharmacists attributed the added value of PGx in being able to determine the cause of an aberrant response to a drug, although they noted to be aware that there will not always be a genetic cause.

Overall, the effects on professional interactions were very positive. Pharmacists felt that GPs perceived them as experts on the subjects and that they respected the initiative they had taken to implement PGx. Pharmacists also felt they could help GPs by taking this task upon them. since GPs were perceived to be too busy for this additional task.

The following enabler corresponds to the TICD Patient factors domain. All pharmacists reported that patient response to PGx was very positive and that they believed in the effects of PGx. A majority of pharmacists also perceived patients to be further interested and therefore were motivated to participate in the PREPARE study to receive their PGx results. Although pharmacists perceived patient interest, they also reported they felt a large portion of patients who did not understand what PGx was and how resolving a PGx interaction would benefit them. Nonetheless, lack of understanding did not prevent their perceived positive effects on the patient–pharmacist relationship. Pharmacists also reported that patients were rarely worried about privacy issues; only one pharmacist reported a patient questioning whether the DNA results would be shared with police officials. Interestingly, pharmacists did take into account which patient preferences were in their decision to adhere to the DPWG recommendation. The majority of pharmacists reported that if a patient were to disagree with a pharmacotherapy adjustment based on PGx test results, this would be a reason not to adhere to the DPWG recommendation. However, no examples of patient disagreement occurred during the study at the time of interview.

## 4. Discussion

This study assessed pharmacists’ perceived remaining barriers preventing, and enablers facilitating, implementation of pharmacist-initiated PGx panel-testing in primary care in the Netherlands, by utilizing a theoretical framework, among pharmacists with experience in handling PGx recommendations. This study is the first in the PGx field to utilize a theoretical framework in exploring determinants of implementation among pharmacists with real-world experience in PGx.

The importance of implementation science has previously been highlighted [38,39]. Frameworks in implementation science are used to identify factors believed to influence implementation outcomes. Determinant frameworks, specifically, are used to identify barriers and enablers of implementation [40]. To date, a number of empirical studies used the TICD and other theoretical frameworks to explore implementation issues across therapeutic areas, including lower back pain [41,42], hand hygiene [43], blood transfusion [44], medication prescribing [45], laboratory testing [46], polypharmacy [47], evidence-based recommendations for chronic conditions [48] and primary care [49], schizophrenia [50] and dementia [51]. In general, uncertainty about how to choose interventions that best match implementation determinants in a given context have been reported [52]. Multiple determinant frameworks may have been suitable for this context, however, we chose to use the TICD framework to address our primary aim through an inductive approach. Utilizing an inductive approach from framework selection allows for the unexpected, and permits more socially-located responses from interviewees [53]. The themes which emerged from the interviews cover all seven domains of the TICD. The majority of the domains emerged both as a barrier and an enabler, supporting the notion that the implementation of PGx is multifactorial and requires a multi-component implementation approach. Interestingly, a domain which only emerged in the enabler themes was patient factors, indicating that the effects of PGx on patients is a strong driver for PGx implementation. In contrast, a number of pharmacists reported that they would not adhere to the DPWG guidelines if patients preferred not to, or had already started using their medication, indicating that DPWG guideline adherence may be hampered by patient shared decision making.

Pharmacists reported unclear procedures outside the study setting as a prominent barrier hindering implementation. Although pharmacists report perceiving themselves as a leading candidate for the selection of patients eligible for PGx, requesting PGx testing, discussing the recommendation with the treating physician and reporting the recommendation, they reported the need for clear guidelines outlining procedures. The mismatch between demand for and lack of practical procedures seems to hamper implementation. No clear solution was provided by the pharmacists, but pharmacists felt that the professional pharmacist organization could contribute to solving this issue. Additionally, the uncertainty on reimbursement, specifically of pharmacist time, was perceived as a strong barrier. Furthermore, even though the nationwide CDSS for PGx in the Netherlands is one of the most advanced in the world, infrastructure inefficiencies were reported as a prominent barrier. This particular barrier may be specific to the pre-emptive panel setting, where PGx results are recorded in the EMR to enable the CDSS in future use. More specifically, pharmacists reported individual DPWG recommendations to be unclear and recording PGx results in the EMR tedious and error-prone. However, the majority of reported recommendations regarded the DPWG recommendations for the *SLCO1B1*-simvastatin and *CYP2D6*-metoprolol interactions. These guidelines, in particular, may be difficult to interpret, due to the stratification of indication or additional risk factors and multiple actions, provided by the recommendation. Therefore, if pharmacists have received recommendations for other gene-drug interactions, which for example are not stratified by indication and only provide one action, they may have reported alternative opinions about guideline clarity. Moreover, pharmacists reported insufficient evidence of clinical utility and cost-effectiveness for pre-emptive PGx panel-testing as a prominent barrier for implementation. In spite of this absence, pharmacists reported to strongly believe in the beneficial effects of PGx, both in the interviews and in the pharmacist survey, where mean reported belief in effect of pre-emptive PGx testing was 4.1 out of 5. Lastly, pharmacists reported a lack of PGx knowledge and awareness among the general HCP population to be hampering implementation. However, completion of the PREPARE study [19] may provide sufficient evidence for both the clinical utility and cost-effectiveness of pre-emptive PGx panel-testing to drive decisions on reimbursement, which may, in turn, provide clarity regarding practical solutions and in turn boost awareness among HCPs.

In contrast to the reported barriers, defined enablers consisted of pharmacist perception of playing a leading role in delivering PGx and the believed beneficial effects of PGx on pharmacotherapy, professional interactions and pharmacist added value and knowledge. Specifically, pharmacist expertise in medication surveillance and a deep understanding of pharmacology were used as arguments to support their leading role. Interestingly, handling PGx interactions was considered similar to other routine pharmacist tasks, such as handling drug-drug interactions and performing medication reviews. Additionally, pharmacists felt handling PGx interactions was of utmost strategic value to illustrate their profession’s added value. Furthermore, a reported strong enabling factor was the positive impact on both professional and patient interactions. The positive impact on professional interactions is further supported by the results of the PGx specific surveys, which show both a high level of discussion and agreement with the resulting decision. Here, 77.2% of PGx results were discussed with the treating physicians (including GPs and medical specialists), with pharmacists being in support of the agreed-upon pharmacotherapy adjustment in 82% of cases. The positive impact on patient interactions as reported in the interviews is also supported by the PGx specific survey, which showed that pharmacists reported the majority (in 56.5%) of PGx recommendations to patients. Interestingly, pharmacists reported the lack of knowledge and awareness of PGx of HCPs outside the scope of the study to be hindering factors, while participating pharmacists self-reported a high level of confidence in their own PGx knowledge. This was reflected both in the interviews and in the surveys, where they rated their knowledge to be 3.3 out of 5, on average. This is remarkable, since only 33.3% reported following the U-PGx e-learning, either completely or partially. This could be explained by the fact that pharmacists reported on-the-job learning by participating in this implementation study. In addition, 33% of participating pharmacists also had experience from an earlier PGx study [28]. This is in accordance with data showing that providers displayed dramatic increases in personal genomic understanding through program participation [54]. Based on this observation, it seems there is less demand for additional PGx education though e-learning, when pharmacists have had hands-on experience with PGx through implementation programs.

Although some studies have investigated barriers of implementation, as reported by physicians with real-world PGx experience [24,25], to our knowledge, only one other study reported on physician perceived barriers to use of PGx, specifically within an implementation study [55]. Although this study was performed in a different healthcare setting, and among physicians as opposed to pharmacists, a number of similar perceived barriers as in physicians were reported, including the lack of clear guidelines for using PGx information and lack of provider knowledge and awareness. Interestingly, the physicians also reported similar enthusiasm towards PGx and process indicators showed a high adoption rate of PGx recommendations in pharmacotherapy.

Since physicians and pharmacists have different backgrounds and responsibilities in patient care, one may expect a discrepancy between their perceived barriers and enablers of PGx, particularly on themes regarding PGx knowledge and awareness and their perceived roles in delivering PGx. While current literature indicates that physicians self-report deficits in PGx knowledge [24,25], this knowledge did improve after having experience with PGx [54]. Similarly, pharmacists in the general population report high PGx awareness, but low PGx knowledge and adoption [17]. On the other hand, in our study, PGx knowledge was self-perceived as sufficient, potentially due to hands-on experience with PGx. In our study pharmacists also perceived to have an important role in the delivery of PGx. Similarly, primary care physicians have also envisioned playing a major role in the delivery of PGx, but recognize their lack of adequate knowledge [56].

Our study has a number of limitations. Firstly, our findings may be prone to selection bias, since this is a study among early-adopting pharmacists who are voluntarily participating in a PGx implementation study, and are therefore more likely to believe in PGx compared to the general pharmacist population. However, a nationwide survey of pharmacists’ perception of PGx showed that 99.7% of the pharmacists believed in the concept of PGx, however, only 14.1% felt adequately knowledgeable [17]. Therefore, our sample may have a similar belief, but higher self-reported PGx knowledge than the general pharmacist population. Secondly, the generalizability of these results for other healthcare settings and other countries may be limited due to differences in IT infrastructure, pharmacist responsibilities, HCP education, reimbursement policies, and patient attitudes. For example, the healthcare IT system in the Netherlands is highly automated and advanced compared to other European countries [57]. Therefore, countries with a less developed CDSS may experience other barriers specific to a panel-approach than reported here. Additionally, in the Netherlands pharmacists are considered healthcare practitioners, and are therefore responsible for the optimization of patient pharmacotherapy. The role of pharmacists may differ in other countries. Both the nature of the reported implementation barriers and their specificity to the healthcare setting, calls for evaluating the implementation process on national or even regional levels. However, a definite advantage of utilizing a systematic approach, by using the TICD theoretical framework, is that findings can readily be compared with implementation in other settings. Lastly, our study has investigated pharmacist-reported determinants of implementation. However, pharmacists are only one of the many stakeholders involved in the implementation of this complex intervention. Therefore, investigation of reported barriers of other stakeholders, such as healthcare payers, other HCPs, professional organizations, and patients may be useful to further tailor a successful implementation strategy.

## 5. Conclusions

Pharmacists-reported enablers of PGx panel-testing include the perceived believed clinical utility of PGx testing and their role in delivering PGx to the patient. Despite these enablers, pharmacists also reported barriers that hinder the implementation in primary care. Pharmacist-reported barriers included unclear procedures outside the study setting, unclear reimbursement for testing and consulting, insufficient evidence of the clinical utility for PGx-panel-testing, insufficient awareness and PGx knowledge among other HCPs, and infrastructure inefficiencies. Knowledge of identified enablers and barriers will support the implementation of routine PGx testing in primary care.

## Figures and Tables

**Figure 1 jcm-09-00814-f001:**
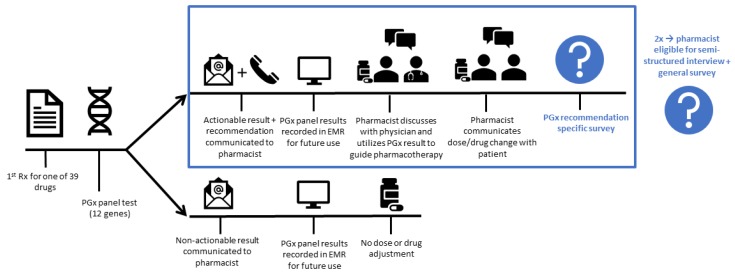
Pharmacist journey per enrolled patient. Pharmacists invite patients who initiate one of the 39 drugs with a Dutch Pharmacogenetics Working Group (DPWG) recommendation. After informed consent, a DNA sample is collected and sent to the Leiden University Medical Center Pharmacogenetics Lab for a pharmacogenomics (PGx) panel test encompassing 12 pharmacogenes. Actionable test results for the drug of enrolment are directly communicated to the pharmacist by phone and email. Other PGx test results are communicated by email. The pharmacist is responsible for recording the PGx results in the electronic medical record (EMR). The pharmacist may choose whether to adhere to the DPWG recommendations. Approval of the treating physician is required before any changes to the drug treatment can be made. Following the report of an actionable PGx result, a PGx recommendation specific survey is performed. Pharmacists were invited for a semi-structured interview and general survey if they had managed at least two patients with actionable PGx results.

**Table 1 jcm-09-00814-t001:** Demographics of interviewed pharmacists, their pharmacies and the drugs, for which actionable recommendations were received before being interviewed.

Characteristics Interviewed Pharmacist (*n* = 15)	
Age (years)	
Mean (SD, range)	38.5 (9.9, 25–59)
Gender	
Female	53.3%
Work experience (years)	
Mean (SD, range)	12.9 (9.0, 0.5–30.0)
Role in pharmacy	
Managing pharmacist	73.3%
Supporting pharmacist	6.7%
Community pharmacy specialist in training	20.0%
Self-reported pharmacogenomic knowledge (scale 1–5)	
Mean (SD, range)	3.3 (0.6, 2–4)
Self-reported belief in effect of pre-emptive pharmacogenomic testing (scale 1–5)	
Mean (SD, range)	4.1 (0.6, 3–5)
Completed PGx e-learning?	
Yes, completely	20%
Yes, partly	13.3%
No	66.7%
When partly, what section of the e-learning did you complete?	
The information videos	13.3%
Number of recommendations received at time of interview	
Mean (SD, range)	3.5 (2.5, 2–11)
Participation in previous PGx study	
Yes	5 (33%)
**Pharmacy characteristics (*n* = 13) ^1^**	
Number of patients	
<5000	7.7%
5000–7500	-
7500–10,000	46.2%
>10,000	46.2%
Full-time equivalents of pharmacists	
≤1 FTE	30.8%
>1 and ≤2 FTE	61.5%
>2 and ≤3 FTE	7.7%
Pharmacy organization	
Independent	23.1%
Franchise	23.1%
Pharmacy group	53.8%
Located in a healthcare center?	
Yes	38.5%
**Top 5 drugs for which actionable recommendations were received by interviewed pharmacists**	
Simvastatin	25 (27.2%)
Metoprolol	22 (23.9%)
Tramadol	11 (12.0%)
Amitriptyline	8 (8.7%)
Atorvastatin	5 (5.4%)

Abbreviations used: SD = standard deviation, FTE = full time equivalent. ^1^ Multiple pharmacists worked in the same pharmacy.

**Table 2 jcm-09-00814-t002:** Shared decision making, report of results to patients, and time allocation, as a result of actionable PGx recommendations (*n* = 92).

Shared Decision Making	
Did you discuss the pharmacogenomic recommendation with the treating physician?	
Yes	77.2%
No	18.5%
Missing	4.3%
Do you agree to the final treatment decision/change that was made based on your PGx-guided recommendation?	
Yes	82.6%
No	2.2%
Missing	15.2%
Why do you disagree? (*n* = 2) “I would have preferred to alter the drug but the patient had already initiated it and therefore I preferred not to alter it”; “The patient was already set on a dose of metoprolol and therefore I preferred not adjusting it”	
**Report of results to the patient**	
Who discussed the results of the PGx recommendation with the patient?	
Pharmacist	56.5%
Pharmacy technician	3.3%
Treating physician	15.2%
Result was to be reported at the time of interview	20.7%
Missing	4.4%
**Time allocation**	
Approximately how much time did you spend on handling this recommendation?	
Mean number of minutes (SD)	18.6 (13.1)
Range	3–90
Missing	10

Abbreviations used: SD = standard deviation, ADEs = adverse drug events.

**Table 3 jcm-09-00814-t003:** Illustrative quotes of emergent barrier themes and their congruent Tailored Implementation for Chronic Diseases (TICD) domains.

Interview Findings: Pharmacist Perceived Remaining Barriers
**(1) Unclear procedures outside the study setting (individual health professional factors; capacity for organizational change)**
“No, I don’t know how to [request a test]. There seems to be some kind of system where you can order tests electronically, but I don’t have access to it anyway.” (P3:16) “Officially, a prescriber still has to request it and that is particularly irritating. I just want to arrange [requesting tests] myself.” (P12:9) “Can the pharmacist also request [PGx tests]? I have to say that the reimbursement policy is really unknown to me.” (P15:23) “... if there was clarity about reimbursement, what does it cost, which patients are eligible—sort of practical guidelines, that would be really useful.” (P14:16) “Requesting a test wasn’t really complicated at all. However, it is still unclear what [genes] to request; do we request the full profile or are you going to request one gene specifically?” (P5:38) “Well I know I can request a gene test in Leiden. If I request in Rotterdam, then the whole panel is tested. But how to make those choices, that is unclear to me.” (P10:30)
**(2) Undetermined reimbursement for test and consult (incentives and resources; social political and legal factors)**
“I don’t mind [the lack of reimbursement] in the experimental phase, but at a certain point, if it becomes more daily practice, then I think there must be something to compensate for [our time].” (P8:43) “If it starts becoming routine practice, then yes, I would think it would be logical to receive compensation for the consultation—that our time is reimbursed by the insurance.” (P9:28) “Well, what I really find a major obstacle is that we are not compensated for the consultations. When I look at how much energy we invest here, we get nothing at all for it. I think that is really a major obstacle because that is not feasible of course.” (P12:39)
**(3) Insufficient evidence of clinical utility for PGx panel-testing (guideline factors)**
“I still think so, yes, research has to show if it is at all cost-effective.” (P1:23) “The insurer is only thinking about cost-benefit ratios. So we should show that its cost-effective or cost-saving so that patients do not receive ineffective means. But of course, we hope to demonstrate that in the PREPARE study. Nonetheless, those [genetic testing] prices really have to really go down.” (P12:21) “We are still implementing in a research context, and investigating its added value. Similarly to implementing a new drug, it has to have demonstrated added value before prescribing it in the clinic. They must first prove that first.” (P5:53)
**(4) Infrastructure inefficiencies (guideline factors; incentives and resources)**
“Well, I think it’s really important that clear and practical guidelines are incorporated into our EMR.” (P15:26) “Not all recommendations are very clearly interpretable.” (P3:2) “The DPWG recommendations really help a lot, even though they are not always very clear. So for example ‘avoid clopidogrel’, well with TIA you do not have many alternatives than clopidogrel, and dipyridamole is unavailable at the moment - sometimes I want the guidelines to be more concrete.” (P15:9) “Well, what I find the biggest obstacle is the limited automation in the pharmacy system.” (P11:37) “The best thing would be if we received the data from the LSP from the lab, of course.” (P13:32)
**(5) Healthcare Professional pharmacogenomics knowledge and awareness (professional interactions; individual health professional factors:)**
“Well, I don’t think it’s very nice to say, but the GPs don’t know anything about it” (P5:15) “I notice that the GPs are not interested in the details, they want to act upon the results but are not interested in anything with CYPs, that’s my perception” (P12:13) “It really depends on the medical specialty, whether [PGx] is of interest to them. For example, the psychiatrists know quite a bit about [PGx], but I know how generalizable this is. On the other hand, I know a patient who was very proud of their PGx profile and showed it to their cardiologist, who had absolutely no idea what it was” (P8:31)

Quote (Pharmacist number: quote number)

**Table 4 jcm-09-00814-t004:** Illustrative quotes of emergent enabler themes and their congruent Tailored Implementation for Chronic Diseases (TICD) domains.

Interview Findings: Pharmacist Perceived Enablers
**(1) Perceived role in delivering PGx (incentives and resources; professional interactions; capacity for organizational change; social political and legal factors)**
Request PGx test “Now and then GPs call me to ask whether requesting a PGx test for particular patients is useful.” (P1:17) “The GPs are just really busy, so I think they appreciate that we take PGx upon us.” (P12:12) “I notice that the GPs really want the [PGx] information but they think it is fine if we request the tests. They have even provided us with a signed empty [requesting] form, and let us fill in what [PGx] tests we need. That has happened twice now.” (P12:14) Acting upon PGx test and reporting to patients “We have a very important role because we should know most about it, at least in primary care.” (P4:37) “[PGx testing] really is the task of the pharmacist because we are in the world of contraindications, interactions, and medication surveillance” (P10:26) “The collaboration [with the GPs] is really good, but they think ‘this has something to do with the liver and can cause intoxications or ineffective plasma levels, you know what‒this is your thing.” (P11:19) “I feel that when I have done all the preparatory work, then its fun to report the results to the patient. Especially when its something simple like “you will be getting another statin.” (P2:11) Follow-up “I feel the follow-up should be a shared responsibility between pharmacist and GP. If the pharmacotherapy has changed as a result of PGx, then both GP and pharmacist should be monitoring how things are going.” (P3:10)
**(2) Believed effects of PGx (individual health professional factors; patient factors)**
Pharmacotherapy improvement “Being able to select those patients at higher risk for adverse drug events before initiating the drug, that is very beneficial” (P13:17) “I think [PGx] may improve drug adherence, I think so.” (P6:25) “I think [PGx] will prevent a lot of healthcare costs related to hospital admissions.” (P15:31) “[testing diagnostically] may not always give a definitive answer, but at least we will be able to cross-out genetics as being the cause [of the adverse event].” (P5:31) “We are now able to fine-tune pharmacotherapy.” (P3:20) “I don’t know if we are saving lives with it, but [PGx] is beneficial and fun.” (P2:50) Pharmacist added value and learning by doing “[PGx] is a great opportunity for pharmacists to show what we can do because a lot of people really don’t know that” (P8:38) “[PGx] makes [pharmacist value] transparent for patients: What does the pharmacist actually do? What is the value of a pharmacy?” (P6:5) “[PGx] gives a really good feeling. This is what I do this profession for.” (P11:15) “This [added value] is the reason why I wanted to participate in this study because I want some experience with PGx testing, for the GPs too” (P7:33) “The more [actionable PGx] interactions you encounter, the easier it becomes.” (P7:20) Professional interaction improvement “[PGx] brings you closer to patients, which really is an added value, and also to the GPs. So I really enjoy doing it.” (P2:56) “[PGx] can give patients a certain feeling of trust in their medication when we say we are going to test your DNA to see if the medication fits your profile, then patients trust it more to start taking it.” (P6:26) “[PGx] confirms what the patient most of the time already knows.” (P1:24)

Quote (Pharmacist number: quote number).

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
