# Peer review of "Assessing the Implementation of Pharmacogenomic Panel-Testing in Primary Care in the Netherlands Utilizing a Theoretical Framework"

_jcm, 2020, doi:10.3390/jcm9030814_

Round 1

Reviewer 1 Report

Thank you for giving me the opportunity to review this manuscript entitled ‘Assessing the implementation of pharmacogenomic panel-testing in primary care in the Netherlands utilizing a theoretical framework’.

PGx testing in clinical implementation associated with germline genes, such as drug-metabolizing enzymes has been delayed around the world, despite precision medicine based on somatic mutation is expected to be widely available as routine clinical practice.

In this manuscript, the authors identified the enablers and barriers for implementation of routine PGx testing in primary care by using both surveys and semi-structured interviews based on theoretical frameworks.

This manuscript has novelty, since there is no qualitative investigation   in this area for community pharmacists so far.

My comments are below

Minor points:

1) The study identified patients planning to start 39 drugs and conducted panel tests on 12 genes as described in the manuscript. To the best of knowledge, the papers related to DPWG have never been published in this journal.

Therefore, please specify 12 genes and/or 39 drugs considered to be clinically implementable to be able to understand for the reader of this journal.

2) The following author’s papers cited in this manuscript have 14 genes and 49 drugs. What is the difference between these targets?

  1. van der Wouden CH et la. Development of the PGx-Passport: A Panel of Actionable Germline Genetic Variants for Pre-Emptive Pharmacogenetic Testing. Clin Pharmacol Ther. 106: 866-873. 2019

3) The importance of PGx education for physicians and pharmacists has been pointed out (Clin Pharmacol Ther. 99: 582-584. 2016, Front Pharmacol. 7: 241. 2016). In this study, only 20% participants completed U-PGx e-learning. Is there any effect on this study results?

4) Throughout this study, demonstrating the clinical utility and cost-effectiveness of gene-panel testing are one of major problem as some pharmacists stated. What about the cost-effectiveness of this panel testing? Please discuss a little more.

5)・P9 Individual health professional factors:

‘Other pharmacists stimulated …………… to educate them om PGx and create  …………….    om ⇒ on  ??

・Additional file 1

  1. Why do you not support this? (when question 4 was answered with b)

   question 4 ⇒ question 3  ??

Author Response

Thank you for giving me the opportunity to review this manuscript entitled ‘Assessing the implementation of pharmacogenomic panel-testing in primary care in the Netherlands utilizing a theoretical framework’.

PGx testing in clinical implementation associated with germline genes, such as drug-metabolizing enzymes has been delayed around the world, despite precision medicine based on somatic mutation is expected to be widely available as routine clinical practice.

In this manuscript, the authors identified the enablers and barriers for implementation of routine PGx testing in primary care by using both surveys and semi-structured interviews based on theoretical frameworks.

This manuscript has novelty, since there is no qualitative investigation   in this area for community pharmacists so far.

My comments are below

Minor points:

1) The study identified patients planning to start 39 drugs and conducted panel tests on 12 genes as described in the manuscript. To the best of knowledge, the papers related to DPWG have never been published in this journal.

Therefore, please specify 12 genes and/or 39 drugs considered to be clinically implementable to be able to understand for the reader of this journal.

We thank the reviewer for this suggestion and have specified the genes and drugs in the newly added Additional file S1 which is referred to in 2.1 Study Setting.

2) The following author’s papers cited in this manuscript have 14 genes and 49 drugs. What is the difference between these targets?

  • van der Wouden CH et la. Development of the PGx-Passport: A Panel of Actionable Germline Genetic Variants for Pre-Emptive Pharmacogenetic Testing. Clin Pharmacol Ther. 106: 866-873. 2019

Indeed, the published PGx-passport as described in reference 15 includes several additional variants located within additional genes (HLA-A and NUDT15). However, these variants and genes have been removed from the panel applied during the course of the PREPARE study for two different reasons. First, variants in HLA-A were excluded due to technical limitations of the utilized genotyping platform in PREPARE. For example, the use of proxy SNPs which were not suitable in Caucasian populations. Second, variants in NUDT15 were not included in PREPARE since the DPWG guideline for the NUDT15-thiopurine interaction was released after PREPARE study initiation. The difference in the number of drugs described in both papers is a result of different panels applied; only drugs potentially interacting with included genes were of interest. To prevent confusion among readers we have mentioned this discrepancy in 2.1 Study Setting:

“After informed consent, a DNA saliva sample is collected in the pharmacy and sent to Leiden University Medical Center Pharmacogenetics Lab for a PGx panel test encompassing 12 pharmacogenes (see Additional File S1). This panel differs from a previously published version of the panel in the PREPARE study (15) since it excludes variants in HLA-A and NUDT15.”

However, we feel the reasoning for the discrepancy is irrelevant for this particular manuscript and have therefore not included it in the revision.

3) The importance of PGx education for physicians and pharmacists has been pointed out (Clin Pharmacol Ther. 99: 582-584. 2016, Front Pharmacol. 7: 241. 2016). In this study, only 20% participants completed U-PGx e-learning. Is there any effect on this study results?

Indeed, only 20% of pharmacists reported fully completing the U-PGx e-learning and another 13.3% reported partly completing it. However, participating pharmacists self-reported a high level of PGx knowledge (3.3/5). Therefore, it appears participating pharmacists developed their PGx knowledge through participation in PREPARE. Additionally, 33% of participating pharmacists had already participated in a previous PGx implementation study (see reference 28) . This rapid gain of PGx experience is in accordance with a previous study where providers displayed dramatic increases in personal genomic understanding through program participation. Based on this observation, it seems there is less demand for additional PGx education though e-learning when pharmacists have had hands-on experience with PGx through implementation programs.

To address the reviewers comment the following section has been added to the discussion:

“This could be explained by the fact that pharmacists reported on-the-job learning by participating in this implementation study. In addition, 33% of participating pharmacists also had experience from an earlier PGx study (28). This in accordance with data showing that providers displayed dramatic increases in personal genomic understanding through program participation (54). Based on this observation, it seems there is less demand for additional PGx education though e-learning when pharmacists have had hands-on experience with PGx through implementation programs.”

4) Throughout this study, demonstrating the clinical utility and cost-effectiveness of gene-panel testing are one of major problem as some pharmacists stated. What about the cost-effectiveness of this panel testing? Please discuss a little more.

Indeed, we agree that demonstrating cost-effectiveness is imperative for large scale implementation of PGx panel testing. One of the aims of the PREPARE study is to determine the cost-effectiveness of the applied panel. As a result of the reviewer’s comment, we have pointed this out in more detail within the discussion:

“Lastly, pharmacists reported a lack of PGx knowledge and awareness among the general HCP population to be hampering implementation. However, completion of the PREPARE study (19) may provide sufficient evidence for both the clinical utility and cost-effectiveness of pre-emptive PGx panel-testing to drive decisions on reimbursement, which may, in turn, provide clarity regarding practical solutions and in turn boost awareness among HCPs.”

5)P9 Individual health professional factors:

‘Other pharmacists stimulated …………… to educate them om PGx and create  …………….    om  on  ??

 We thank the reviewer for spotting this typo. We have corrected it.

Additional file 1

6) Why do you not support this? (when question 4 was answered with b)

   question 4 question 3  ??

 We thank the reviewer for spotting this typo. We have corrected it.

Reviewer 2 Report

The manuscript describes pharmacists' perceived barriers and enablers for PGx testing, focusing on pharmacists involved in a trial of preemptive PGx testing. Findings add to the growing body of literature on barriers and facilitators of PGx implementation in clinical care. The manuscript would benefit from comparing/contrasting pharmacist and physician perceived barriers/enablers, especially for themes on HCP PGx knowledge and awareness and perceived roles in delivering PGx.  Otherwise, I have only minor comments:

  1. Phenotypes were recorded as contraindications. It would be helpful to expand on this and whether it applies to all phenotypes.  For example, would a normal metabolizer phenotype be recorded as a contraindication?  Or is this just reserved for non-normal phenotypes. In addition, a contraindication generally means that a drug absolutely should not be used unless the benefits clearly outweigh risks.  In the context of PGx testing, this is often not the case. Rather, the drug dose may be adjusted for poor, intermediate, or ultra-rapid metabolizers. Some discussion on this may be warranted.
  2. The authors may wish to add some discussion about how barriers to preemptive testing may differ from those with reactive testing where turn around time is especially important.
  3. Tables 3 and 4 could be moved to the supplement. 

Author Response

The manuscript describes pharmacists' perceived barriers and enablers for PGx testing, focusing on pharmacists involved in a trial of preemptive PGx testing. Findings add to the growing body of literature on barriers and facilitators of PGx implementation in clinical care. The manuscript would benefit from comparing/contrasting pharmacist and physician perceived barriers/enablers, especially for themes on HCP PGx knowledge and awareness and perceived roles in delivering PGx. 

We thank the reviewer for this valuable suggestion and have added the following lines describing contrasting pharmacist and physician perceived barriers/enablers to the discussion:

“           Since physicians and pharmacists have different backgrounds and responsibilities in patient care, one may expect a discrepancy between their perceived barriers and enablers of PGx, particularly on themes regarding PGx knowledge and awareness and their perceived roles in delivering PGx. While current literature indicates that physicians self-report deficits in PGx knowledge (24, 25), this knowledge did improve after having experience with PGx (54). Similarly, pharmacists in the general population report high PGx awareness, but low PGx knowledge and adoption (17). On the other hand, in our study, PGx knowledge was self-perceived as sufficient, potentially due to hands-on experience with PGx. In our study pharmacists also perceived to have an important role in the delivery of PGx. Similarly, primary care physicians have also envision to play a major role in the delivery of PGx, but recognize their lack of adequate knowledge (56).”

Otherwise, I have only minor comments:

  • Phenotypes were recorded as contraindications. It would be helpful to expand on this and whether it applies to all phenotypes.  For example, would a normal metabolizer phenotype be recorded as a contraindication?  Or is this just reserved for non-normal phenotypes. In addition, a contraindication generally means that a drug absolutely should not be used unless the benefits clearly outweigh risks.  In the context of PGx testing, this is often not the case. Rather, the drug dose may be adjusted for poor, intermediate, or ultra-rapid metabolizers. Some discussion on this may be warranted.

Indeed, the pharmacist are encouraged to record all tested genotype predicted phenotypes as contra-indications, regardless of being aberrant or normal. Reason for this being to record that PGx testing has been performed for a particular gene. To further clarify this for the reader, we have explained this issue in 2.1 Study Setting:

“Pharmacists are encouraged to record all results, regardless of the phenotype being aberrant or normal, to document testing for that particular gene has been performed.”

We understand the reviewer’s discussion whether the term contra-indication is suitable in the PGx scenario. However, in this case, the term contra-indication only relates to the format in which it should be recorded in the EMR and does not imply that the PGx result should be considered as contra-indications. To further clarify this for the reader, we have explained this issue in 2.1 Study Setting:

“Although the term contra-indication generally indicates a particular drug should not be used, this is not implied in the PGx scenario. The term contra-indication only relates to the format in which the genotype predicted phenotypes should be recorded in the EMR to trigger pop-up messages with PGx information during drug prescribing and dispensing.”

  • The authors may wish to add some discussion about how barriers to preemptive testing may differ from those with reactive testing where turn around time is especially important.

The reviewer raises a fundamental issue. Indeed, the barriers reported by HCPs may actually differ when delivered in reactive or pre-emptive PGx models. We have therefore stated this discrepancy in the introduction and further elaborate on its implications in the discussion by identifying barriers which are specific to a panel-approach:

Introduction: “Furthermore, remaining barriers may differ when delivered in a single-gene reactive approach or a pre-emptive panel approach.”

Discussion:

“Furthermore, even though the nation-wide CDSS for PGx in the Netherlands is one of the most advanced in the world, infrastructure inefficiencies were reported as a prominent barrier. This particular barrier may be specific to the pre-emptive panel setting where PGx results are recorded in the EMR to enable the CDSS in future use.”

“Therefore, countries with a less developed CDSS may experience other barriers specific to a panel-approach than reported here.”

However, we did not include the turn-around time as an advantage of a panel approach in the discussion since the applied turn-around time in our study is similar to that of a reactive approach for the drug of enrolment.

  • Tables 3 and 4 could be moved to the supplement. 

We thank the reviewer for their suggestion. However, we feel the quotes included in Table 3 and 4 very much illustrate the identified barriers and enablers and therefore feel it is substantiated to include them in the main tex.t.